genomics, ecology, immunology

ecological genetics, *Syncerus caffer*, ecoimmunology, *Mycobacterium bovis*

**Author for correspondence:**
Hannah F. Tavalire
e-mail: tavalire@uoregon.edu

†Present address: Prevention Science Institute, University of Oregon, Eugene, OR, USA.
‡Present address: Institute of Ecology and Evolution, University of Oregon, Eugene, OR, USA.
¶Present address: SANPARKS, Veterinary Wildlife Services, Skukuza, South Africa.

# Risk alleles for tuberculosis infection associate with reduced immune reactivity in a wild mammalian host

Hannah F. Tavalire[1,†,‡], Eileen G. Hoal[3], Nikki le Roex[3,¶], Paul D. van Helden[3], Vanessa O. Ezenwa[4] and Anna E. Jolles[1,2]

[1]Department of Integrative Biology, and [2]College of Veterinary Medicine, Oregon State University, Corvallis, OR, USA
[3]South African Medical Research Council, DST/NRF Centre of Excellence for Biomedical TB Research, Division of Molecular Biology and Human Genetics, Faculty of Health Sciences, Stellenbosch University, Tygerberg, South Africa
[4]Odum School of Ecology and Department of Infectious Diseases, College of Veterinary Medicine, University of Georgia, Athens, GA, USA

(iD) HFT, 0000-0002-5235-0008; VOE, 0000-0002-8078-1913

Integrating biological processes across scales remains a central challenge in disease ecology. Genetic variation drives differences in host immune responses, which, along with environmental factors, generates temporal and spatial infection patterns in natural populations that epidemiologists seek to predict and control. However, genetics and immunology are typically studied in model systems, whereas population-level patterns of infection status and susceptibility are uniquely observable in nature. Despite obvious causal connections, organizational scales from genes to host outcomes to population patterns are rarely linked explicitly. Here we identify two loci near genes involved in macrophage (phagocyte) activation and pathogen degradation that additively increase risk of bovine tuberculosis infection by up to ninefold in wild African buffalo. Furthermore, we observe genotype-specific variation in IL-12 production indicative of variation in macrophage activation. Here, we provide measurable differences in infection resistance at multiple scales by characterizing the genetic and inflammatory variation driving patterns of infection in a wild mammal.

## 1. Background

To predict and control the spread of infections in host populations, we must characterize how different hosts contribute to transmission. Heterogeneity in infection risk and morbidity is present across many disease systems, driving patterns in population-level disease dynamics. For example, tolerant (i.e. individuals able to limit damage at high parasite burdens [1]) though highly susceptible 'superspreaders' create heterogeneity in infection incidence and exposure risk, driving patterns in infection dynamics at the population level in multiple disease systems [2]. In less extreme cases, cryptic genetic variation for infection resistance (i.e. a host's ability to prevent or delay infection) still drives patterns in prevalence and risk that otherwise remain unexplained, especially in natural populations [3,4]. To improve outcomes for individual hosts, we must determine the immunologic and genetic mechanisms underlying variation in infection risk and morbidity. As such, linking scales of organization—from genes to cells to individual hosts to populations and metapopulations—is a central challenge in understanding infectious disease processes.

Bridging the divide between laboratory-based approaches in genetics and immunology and investigations of infection dynamics in natural populations

has been identified as a priority in recent work [5–8]. Nonetheless, multiscale studies that explicitly link genetic and immunological mechanisms to variation in infection in natural populations are exceedingly rare (however, see work in rodents [9–12]). Consequently, even for important and well-studied infectious diseases, the extent to which immunologic and genetic mechanisms cause relevant variation in disease transmission in outbred host populations experiencing the full range of natural environmental variability is usually unknown. Yet interactions among hosts and their parasites can be strong drivers of evolution, and natural variation in immune response is expected to be a frequent target of selection [13–15], and have direct implications for the host's fitness and disease transmission [16–18]. Understanding the genetic and immunological basis for these host defence mechanisms is paramount to disentangling complex coevolutionary dynamics between hosts and their parasites.

Members of the *Mycobacterium* genus are well represented in the list of major challenges to human, livestock and wildlife health [19]. A quarter of the world's human population is plagued with a mycobacterial infection, with 10 million new cases annually [20,21]. *Mycobacterium bovis*, the causative agent of bovine tuberculosis (bTB), is a broad host range zoonotic pathogen capable of infecting most mammals. Though human tuberculosis cases were historically solely attributed to *Mycobacterium tuberculosis* infections, roughly 3% of cases now arise from *M. bovis* infections worldwide (though this is probably still an underestimate due to imprecision of current diagnostic methods [21,22]). *Mycobacterium bovis* infections in livestock also represent a substantial economic burden in most endemic countries, where test-cull control practices or animal morbidity and mortality lead to rampant agricultural losses [21]. Furthermore, *M. bovis* infections in wildlife have been reported in many countries with documented spillover events from and into livestock, and prove to be almost impossible to eradicate from free-ranging maintenance host populations [23–25].

High worldwide disease burden has made tuberculosis a main focus of biomedical research over the last 50 years. Extensive laboratory and clinical studies characterize immune interactions at the cellular level between invading mycobacteria and their hosts. Though innate recognition mechanisms of the host are based on highly conserved pathogen-associated proteins (which are often integral to pathogen survival), multiple species in the *Mycobacterium* genus have evolved complex mechanisms to avoid detection [26–28]. Initially, mycobacterial pathogens are identified by macrophages (a type of phagocyte) and phagocytized (engulfed). Recognition should result in macrophage activation and production of pro-inflammatory cytokines or apoptosis in an effort to control spread, but mycobacteria often interfere with activation of host immune cells [29] and disrupt downstream processes of immune containment [30–32]. Thus, mycobacterial infection often manifests as a reduction in phagocyte activation and resulting pro-inflammatory cytokine signalling, specifically interleukin-12 (IL-12) expression [33,34].

The capacity of *Mycobacterium* species to evade host immune recognition or degradation makes them highly effective pathogens. However, variation in host resistance has not been described outside of laboratory animals and clinical work in humans. Furthermore, though immune evasion is common across the *Mycobacterium* genus, underlying physiological and genetic mechanisms maintaining variation in mycobacterial infection resistance in natural populations have not been described, and the implications of such variation for transmission dynamics are unknown.

Here, we examine the genetic basis for *M. bovis* resistance in African buffalo (*Syncerus caffer*) and relate genotypic variation at candidate resistance loci to variation in cytokine production and infection risk in a natural mammalian host population. African buffalo serve as a maintenance host for *M. bovis* in the savannah ecosystem, sustaining relatively high levels of infection in some areas (up to 27%) and acting as a source of infection for other wildlife and livestock bordering wildlife areas [24,35,36]. Previous work in this disease system has identified multiple distinct forms of host resistance and provides weak evidence that infection resistance may be heritable [37]. Though African buffalo herds can sustain high bTB prevalence, infection resistance in this system operates on a continuum, with some animals succumbing to infection early in life (from 2 to 6 years old) but surviving longer once infected, and more resistant animals preventing infection until later in life, but dying shortly after infection (up to 14 years old [37]). Furthermore, incidence (rate of new infections) is stable across age groups after 4 years of age [37]. These age-dependent infection patterns result in high variation in resistance at the individual level, but relatively stable infection dynamics at the population level. Furthermore, *M. bovis* has been shown in previous studies to alter infection patterns of co-infecting pathogens [38,39] and dramatically impact host fitness in African buffalo [37,40], making *M. bovis* a strong potential driver of evolution in this system. Therefore, African buffalo serve as an ideal wild model system in which to investigate the genetic and mechanistic basis for *M. bovis* infection resistance within a natural setting, shedding light on tuberculosis infection dynamics outside the laboratory.

## 2. Methods

### (a) Study area and field data collection

Two hundred sub-adult and young adult female African buffalo (initial ages 2–7 years) were captured every six months in the southern part of Kruger National Park, South Africa between June 2008 and August 2012 as part of an experiment evaluating the consequences of anthelminthic treatment for bTB transmission (for more detail, see Ezenwa & Jolles [41]). Buffalo were sampled from two distinct herds occurring in the Crocodile Bridge and Lower Sabie areas of the park (35 km apart). Estimated herd sizes during the study period for Crocodile Bridge and Lower Sabie were 2100 and 1100 buffalo, respectively.

Each buffalo was fitted with either a radio (*n* = 193) or satellite (*n* = 7) collar with a high-frequency VHF transmitter upon first capture, that was then used to locate the animal for subsequent captures at roughly six-month intervals. Individuals lost to death or emigration during the study period were replaced to maintain a constant sample size of 200 animals spread equally across the two herds. Of these animals, half (*n* = 50 per herd, *n* = 100 total) were randomly chosen to receive an anthelminthic bolus (slow-release fenbendazole; Panacur, Hoechst Roussel) as part of the study design outlined in Ezenwa & Jolles [41]. Animal age was determined in young animals by tooth emergence and in older animals by wear pattern [42].

At each capture, animals were immobilized by dart from a helicopter or truck using etorphine hydrochloride (M99, Captivon, Karino, South Africa). Following data collection,

immobilization was reversed using diprenorphine (M5050) or naltrexone (40 mg ml$^{-1}$, Kyron). Animals were observed until fully recovered and all immobilizations were conducted by a veterinarian according to the South African National Parks Standard Operating Procedures for the Capture, Transportation, and Maintenance in Holding Facilities of Wildlife. All animal work for this study was approved by the Institutional Animal Care and Use Committee (IACUC) at both Oregon State University (ACUP #3267) and the University of Georgia (UGA No. A201010190-A1), which follow the 8th edition of the *Guide for the Care and Use of Laboratory Animals* [43], the *Guide for the Care and Use of Agricultural Animals in Research and Teaching* [44], and the European Convention for the Protection of Vertebrate Animals Used for Experimental and Other Scientific Purposes [45].

To more accurately assess longitudinal patterns in infection and immune function, we excluded animals with fewer than four capture timepoints.

## (b) Bovine tuberculosis testing and cytokine stimulations

All blood samples for disease diagnostics and cytokine stimulations were collected from the jugular vein within 15 min of sedation and stored in heparinized tubes on ice until processing that day. bTB infection was determined with the commercially available whole-blood gamma interferon (IFNγ) assay (BOVIGAM, Prionics, Switzerland). This assay measures the difference in IFNγ production of whole blood in response to incubation with bovine versus avian tuberculin antigens, while controlling for background IFNγ levels [46]. Individual samples were called as bTB-positive or -negative based on absorbance thresholds optimized for African buffalo [47]. We obtained a time series of 2–9 bTB tests for each animal across 10 total possible captures and used the full time series to more confidently assign bTB status [41]. Animals with at least two consecutively positive bTB tests were assigned as bTB-positive. We excluded animals with alternating bTB test results, as they could not be confidently phenotyped. Since bTB is chronic in buffalo, and there is no evidence of recovery, we assumed animals remain bTB-positive until death [48]. Prevalence of bTB in this sample was 0.142 at the beginning of the study (95% CI (0.090, 0.193)).

At each capture, we assessed host immune function by measuring production of the cytokines interferon gamma (IFNγ), interleukin 12 (IL-12) and interleukin 4 (IL-4). Across mammals, increased levels of IFNγ would be consistent with a T-helper 1 (T$_H$1)-dominated adaptive immune response targeting intracellular pathogens (e.g. viruses and some bacteria), while increased production of IL-4 is commonly associated with a T-helper 2 (T$_H$2) dominated immune response against extracellular pathogens (e.g. helminths [49]). IL-12 production correlates with innate phagocyte activation during an initial, non-specific immune response [49,50] and also plays a critical role in coordinating a successful adaptive T-cell response [51]. Additionally, IFNγ and IL-12 have been identified as key cytokines involved in an effective immune response to tuberculosis infection [33,52].

We used cytokine-specific enzyme-linked immunosorbent assays (ELISAs; Abd Serotec) to quantify levels of each cytokine (IFNγ, IL-12 and IL-4) following stimulation with pokeweed mitogen (*Phytolacca americana*; Sigma). Pokeweed is an established mitogen that elicits both B- and T-cell-dependent immune responses in peripheral blood samples *in vitro*, allowing the assessment of the reactive potential of whole blood immune cells to a novel antigen by measuring cytokine production [53–55]. Detailed cytokine quantification methods for this herd have been previously described (IFNγ [41], IL-12 [56] and IL-4 [57]). Briefly, 1.5 ml aliquots of whole blood from each animal were incubated with 0.3 mg ml$^{-1}$ of pokeweed for 24 h at 37°C,

after which samples were centrifuged, and the plasma supernatant was removed and stored at −20°C until ELISA testing in duplicate. OD-based quantification methods were used to estimate stimulation-associated levels of cytokine production as a proxy of general immune reactivity to a novel stimulus. We obtained a time series of six consecutive measures of IL-12 and 10 consecutive measures of IFNγ and IL-4, which were more intensely sampled based on the priorities of the original study [41]. Animals with less than four samples for IL-12 and less than eight samples for IFNγ and IL-4 were not included in the cytokine analyses. Missing cytokine measures were due to low blood volume collected at capture, or high discrepancy between ELISA replicates.

## (c) SNP genotyping and filtering

We used single-nucleotide polymorphism (SNP)-based molecular methods to identify variable regions of the African buffalo genome for population structure analysis and genome-wide association study (GWAS). We extracted DNA from dried ear tissue samples and prepared individual DNA libraries for sequencing using type IIB restriction-site-associated DNA (2bRAD) methods, detailed in Wang *et al.* [58]. SNP identification, mapping and general quality filtering methods for these data have been previously described [37]. For the GWAS analyses, animals that were genotyped at 5000 or fewer loci were removed from the dataset ($n = 4$). We retained SNP markers that had at least 10× coverage and were genotyped in 90% of individuals. Markers were discarded if they had more than two alleles, violated Hardy Weinberg equilibrium ($p < 0.0001$) or had a minor allele frequency less than 0.05.

To assess independence of each marker, we quantified linkage disequilibrium (LD) among all pairwise combinations of SNPs using $r^2$ and removed markers in high LD. Of the SNPs tested, 15 pairs were in significant LD ($r^2 > 0.5$), 13 of which were within 100 kb on the same scaffold. We removed the four SNPs in high LD that were not physically linked, since these markers are non-independent. Ultimately, filtering yielded 187 usable buffalo samples genotyped at 1480 SNPs.

## (d) Statistics: genome-wide association study

Here, we evaluated bTB infection resistance in African buffalo as variation in time to onset of infection (i.e. conversion age). Since *per capita* incidence was previously shown to be equal among herds [41] and there is weak evidence for marginally heritable variation in infection resistance in this group of buffalo [37], we assume some underlying physiological mechanism with a genetic basis is causing variation in resistance in this system. One would expect stochastic variation in time to infection due to variation in exposure, but on average, more resistant animals should become bTB-positive later in life.

We used right-censored Cox proportional hazards regression models to identify SNPs associating with variation in age at onset of bTB, and therefore variation in infection resistance. Buffalo that converted during the study or never converted to bTB-positive were included in the analysis ($n = 160$), while animals that were bTB-positive at first capture were excluded since their exact conversion age could not be determined. We tested associations of each SNP genotype and allele separately using the R package *survival* [59]. SNP models for time to onset of bTB conversion (conversion age) included genotype or allele as categorical main effects, as well as anthelminthic treatment, herd and initial age as covariates. Anthelminthic treatment was previously demonstrated to impact survival following bTB infection in a subset of this population [41]. Previous work has also demonstrated the two herds described here are not genetically distinct [37,60]. We therefore include the covariate 'herd' to account for large-scale environmental

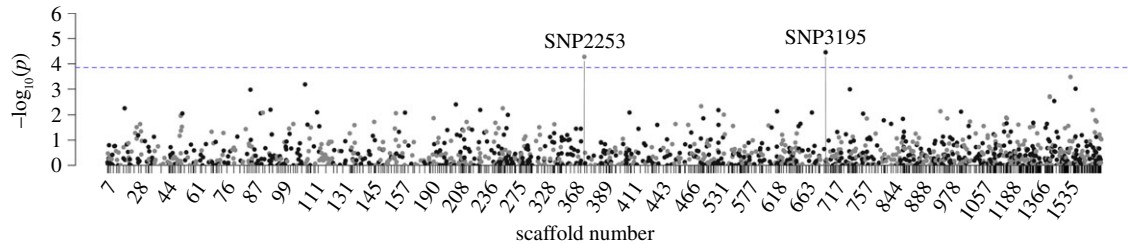

**Figure 1.** Association of SNP genotype in models for time to onset of bTB. The 2000 largest scaffolds are pictured, ordered by size and alternate black and grey in colour. The blue line denotes the significance thresholds for the FDR corrected $q = 0.05$ and significant SNPs are labelled.

variation and not any underlying genetic differences. Finally, initial age was included to account for differences in observation period during the life of the animal (age at first capture ranged from 2 to 7 years). To control for genetic substructure in our sample, we performed a principle component analysis (PCA) using SNP markers in the R package *adegenet* [61] and included the first five axes of this relatedness PCA as fixed effects in our models (as most animals were only distantly related, these components explained 10% of total genetic variation [62]). Collinearity among fixed effects was assessed by calculating variance inflation factors (VIFs) in the R package *rms* [63], however, no instances of collinearity were identified (all VIFs were < 10). SNP $p$-values from each model were false discovery rate (FDR) corrected using the Benjamini–Hochberg procedure to generate adjusted $q$-values based on the original $p$-value distribution [64]. SNPs with adjusted $q$-values less than 0.05 were considered to significantly associate with conversion age. To test for additive or interactive effects of multiple significant SNPs within a multi-locus genotype, we then included genotype at each significantly associating locus in a single Cox regression model. We then used step-wise Akaike information criterion-based model selection [65] from the initial full multi-locus Cox regression model which included the first five PCA axes, anthelminthic treatment, herd, year of first capture and initial age as covariates.

We calculated mean linkage block size to determine an appropriate window around each SNP within which to search for putative candidate genes. We calculated $r^2$-values for all physically linked pairwise SNPs that occurred within 100 kb on the same scaffold, then determined the mean distance at which $r^2$ was greater than 0.9, and therefore reflected the average linkage block size for markers in high LD. We then used this linkage block as a sliding window for candidate gene discovery near significantly associating SNPs. We identified putative bTB resistance candidate genes using the *S. caffer* genome annotation [66]. Any large areas without annotation adjacent to SNPs of interest were further interrogated using InterProScan, which combines multiple methods of protein signature recognition to identify putative full and partial protein coding regions [67,68]. Putative protein regions identified by InterProScan with an associated e-value less than $e^{-50}$ were considered to have sufficiently high assignment confidence.

### (e) Statistics: inflammatory phenotypes

Here, we define variation in 'inflammatory phenotype' as any measurable difference in cytokine production among individuals indicative of variation in innate or adaptive immune activation. IFNγ and IL-12 are important signalling cytokines during the $T_H1$ and phagocyte response to invading *Mycobacterium* spp., respectively [33], while IL-4 indicates a $T_H2$-dominated adaptive response more characteristic of a macroparasite infection [49]. We compared SNP genotype at loci significantly associating with conversion age after FDR correction to IFNγ, IL-4 and IL-12 production over time to determine if variation in immune reactivity associates with bTB infection resistance genotype. Mixed-effects

models were run in *nlme* [69]. Initial full models for each cytokine included animal age in years, herd, anthelminthic treatment, SNP genotype at each significant SNP, bTB status, season, year, the interaction of each SNP with bTB status and the interaction of season with year and animal ID as a random effect. The interaction of bTB status and SNP genotype was initially included in each model as cytokine phenotype may vary with infection. Interactions of season and year were included to allow for inter-annual variation and seasonal patterns in this highly dynamic ecosystem. To control for variation in cytokine level due to variation in plasma storage time and ELISA plate effects, we included plate as a random effect in each corresponding mixed-effects model. Additionally, to control for differences in cytokine measurement number, individual animals were weighted by total number of captures in each model to control for differences in observation period. No instances of collinearity were identified among fixed effects (all VIFs were less than 10).

If SNP genotype is indicative of variation in phagocytic response, we would expect that IL-12 production would differ among SNP genotypes since activated phagocytes release IL-12 [70]. By contrast, we would expect no difference in IFNγ or IL-4 production among genotypes, since these cytokines are a proxy for $T_H1$ and $T_H2$ immune function, respectively, and can be produced independently of phagocyte activation [71].

When applicable, normality was assessed using data visualization and Shapiro–Wilk tests. Cytokines were natural log transformed to meet assumptions of normality. All cytokine model selection was done using the Akaike information criterion [65], conditional $R^2$, and residual plot visualization to test for homoscedasticity [72]. All statistics were run in R version 3.2.4 [73].

## 3. Results

### (a) Genome-wide association

Observed and expected heterozygosity for each of the 1480 SNPs ranged from 0.0604 to 0.5839 (median 0.2177) and 0.0950 to 0.500 (median 0.2188), respectively. Principal components analysis (PCA) revealed no obvious clustering among individuals of each herd that would be indicative of distinct genetic groupings, though some low-level genetic substructure is present (electronic supplementary material, figure S1).

GWAS yielded support for strong associations between age at onset of bTB and genotypic variation at two SNPs on two separate scaffolds (figure 1). These two SNPs remained highly significant ($q < 0.05$) after FDR correction in models with genotype as a main effect, after controlling for background genetic structure (table 1, figure 1). We found the rare allele at each of these loci conferred an additive 5.386 (SNP2253; 95% CI (2.5878, 11.2117)) and 4.084 (SNP3195; 95% CI (2.1823, 7.6436))-fold increase in risk of

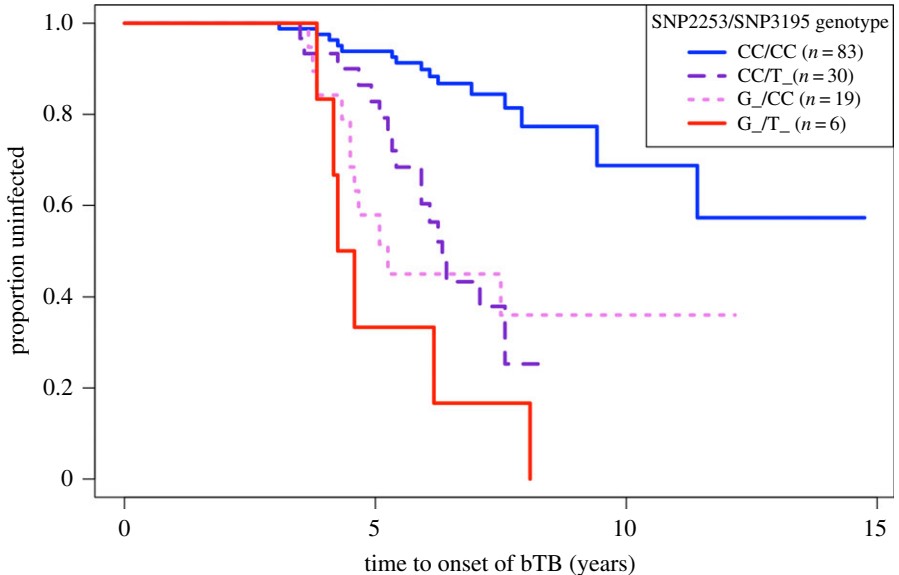

**Figure 2.** Time to onset of bTB by multi-locus genotype. The Kaplan–Meyer time-to-event curve by multi-locus SNP genotype at SNP2253 and SNP3195. Legend denotes each multi-locus genotype and sample size. Across both loci, rare alleles confer a ninefold additive risk of bTB conversion (bottom solid line). (Online version in colour.)

**Table 1.** Cox proportional hazards regression model for time to onset of bTB infection including SNP genotype at the two significantly associated SNPs.

| model | estimate (s.e.)[a] | z-value | p-value |
|---|---|---|---|
| time to onset of bTB | | | |
| PC1 | 0.990 (0.032) | −0.319 | 0.7500 |
| PC2 | 1.001 (0.025) | 0.056 | 0.9556 |
| PC3 | 0.988 (0.032) | −0.366 | 0.7143 |
| PC4 | 1.008 (0.036) | 0.235 | 0.8142 |
| PC5 | 1.083 (0.037) | 2.128 | 0.0333 |
| herd (Lower Sabie) | 1.579 (0.316) | 1.444 | 0.1489 |
| initial age (years) | 0.557 (0.140) | −4.183 | <0.0001 |
| SNP2253 genotype (G_) | 5.386 (0.374) | 4.502 | <0.0001 |
| SNP3195 genotype (T_) | 4.084 (0.320) | 4.400 | <0.0001 |
| $n = 138$; events = 50 | | | $R^2 = 0.371$ |

[a]All estimates are back-transformed and represent a multiplicative increase in risk of bTB conversion.

converting to bTB-positive in this population of buffalo, and no interactive effects were detected in the multi-locus genotype Cox regression (table 1, figure 2). Genotypic frequencies for each significantly associating locus are reported in table 2. Due to low frequency of the risk allele at both loci and a small number of risk allele homozygotes at each locus, we present only the allele model (the genotype model estimated similar conversion risks for heterozygotes and risk allele homozygotes at these loci). These loci were not in LD, thus associations of variation at these loci with bTB conversion risk are likely driven by independent underlying physiological or genetic mechanisms.

We determined the average linkage block size for markers in high LD ($r^2 \geq 0.9$) to be 29 kb and therefore used this distance to guide our search for genes near each SNP that are potentially involved in underlying infection resistance mechanisms. Within 29 kb of SNP2253, we found one gene: peroxisomal membrane protein *PEX14* (18 kb downstream).

**Table 2.** Genotype frequencies at each SNP significantly associated with time to onset of bTB infection.

| locus | genotype | frequency (n) |
|---|---|---|
| SNP2253 | CC | 0.810 (119) |
| | CG | 0.184 (27) |
| | GG | 0.006 (1) |
| SNP3195 | CC | 0.735 (111) |
| | CT | 0.245 (37) |
| | TT | 0.020 (3) |

We failed to identify any genes within 29 kb of SNP3195 using the *S. caffer* genome annotation and only identified small coding region fragments (less than 50 bp) using InterProScan.

**Table 3.** Mixed-effects maximum-likelihood models for longitudinal production of each cytokine by the presence–absence of the SNP2253 risk allele (G).

| model[a] | estimate (s.e.)[b] | t-value | p-value |
| --- | --- | --- | --- |
| (a) IL-12 production (pg ml$^{-1}$) | 402.917 (0.144) | 41.770 | <0.0001 |
| herd (Lower Sabie) | 0.483 (0.157) | −4.623 | <0.0001 |
| SNP2253 (G_) | 0.551 (0.187) | −3.187 | 0.0023 |
| season (wet) | 0.772 (0.152) | −1.697 | 0.0915 |
| | | | n = 64$^c$; 15 w/risk allele |
| (b) IFNγ production (ng ml$^{-1}$) | 0.636 (0.074) | −6.147 | <0.0001 |
| treatment (control) | 0.807 (0.099) | −2.160 | 0.0349 |
| SNP2253 (G_) | 0.818 (0.148) | −1.357 | 0.1800 |
| bTB (+) | 1.142 (0.096) | 1.378 | 0.2621 |
| SNP2253 (G_) × bTB (+) | 1.407 (0.016) | 2.115 | 0.1247 |
| | | | n = 62$^c$; 11 w/risk allele |
| (c) IL-4 production (pg ml$^{-1}$) | <0.001 (201.564) | −1.653 | 0.0991 |
| SNP2253 (G_) | 0.461 (0.433) | −1.787 | 0.0790 |
| bTB (+) | 0.701 (0.339) | −1.046 | 0.4856 |
| capture year | 1.184 (0.100) | 1.683 | 0.3414 |
| season (wet) | 4.385 (0.231) | −3.573 | 0.1737 |
| SNP2253 (G_) × bTB (+) | 7.154 (0.641) | 3.069 | 0.2005 |
| | | | n = 62$^c$; 11 w/risk allele |

[a]Production of each cytokine was measured following incubation of whole blood with a pokeweed mitogen.
[b]Estimates in all cytokine models are back-transformed and represent a multiplicative increase in cytokine production.
[c]Only animals with one missing data point or less are included in each model to control for consistency of age distribution across samples (64 animals had at least 4/5 samples for IL-12 and 62 animals had at least 8/9 samples for IL-4 and IFNγ).

## (b) Inflammatory phenotypes

We assessed broad-scale putative mechanisms of increased bTB conversion risk in this herd by comparing allelic variation at the loci of interest to cytokine production following whole blood stimulation. Animals heterozygous or homozygous for the risk allele (G_) at SNP2253 produced 45% less IL-12 than animals with the CC genotype regardless of bTB status, while allelic variation at SNP2253 was not predictive of differences in IFNγ or IL-4 production (table 3, figure 3). These results suggest no difference in activation of $T_H1$ and $T_H2$ cells among SNP2253 genotypes, but that the production of IL-12 from activated phagocytes in the blood (monocytes/macrophages) may be reduced in animals harbouring the risk allele. By contrast, we observed no significant patterns in cytokine production relative to SNP3195 alleles and this factor was not retained during model selection in any of the cytokine models. We therefore found no cytokine-based evidence for variation in $T_H1$, $T_H2$ or phagocyte activation associating with variation in SNP3195 genotype (electronic supplementary material, figure S2). Taken together, these results suggest distinct mechanisms of infection resistance associating with variation at these two loci that have an additive effect on overall infection risk (figure 3).

## 4. Discussion

Here, we demonstrate genetic variation near a specific gene related to anti-bacterial immunity and phagocyte activation associating with measurable differences in immune reactivity and driving massive variation in bTB infection resistance in a wild mammalian population. This is one of very few studies taking a multi-level approach, connecting infection resistance phenotypes across scales, from an individual-level disease trait to patterns in immune reactivity in the wild. Furthermore, one genomic region associating with age at onset of bTB contains the gene *PEX14* which is directly related to inducible anti-bacterial immunity and apoptosis of infected phagocytes [74,75]. Production of IL-12 during whole blood stimulation is significantly reduced in animals harbouring the risk allele at this locus and these animals are at an over fivefold increased risk of succumbing to bTB. Taken together, these putative candidate mechanisms and variation in IL-12 relative to genotype suggest appreciable differences in inflammatory phenotype among buffalo of different genotypes at SNP2253, which directly impacts risk of bTB infection. By contrast, we detected no association between SNP3195 genotype and the cytokines assayed here, suggesting that alternative immune mechanisms underlie variation in bTB risk associating with this locus. We were also not able to further annotate the 'gene desert' surrounding SNP3195, and thus cannot put forth putative mechanisms for increased infection risk associating with the rare allele at this locus.

Multiscale studies of infectious diseases in wild populations can yield discoveries in applied and basic aspects of infectious disease biology, and build upon laboratory-based studies. For example, foundational ecoimmunology work in wild rodents has uncovered dramatic differences in immune function between laboratory mice and their wild

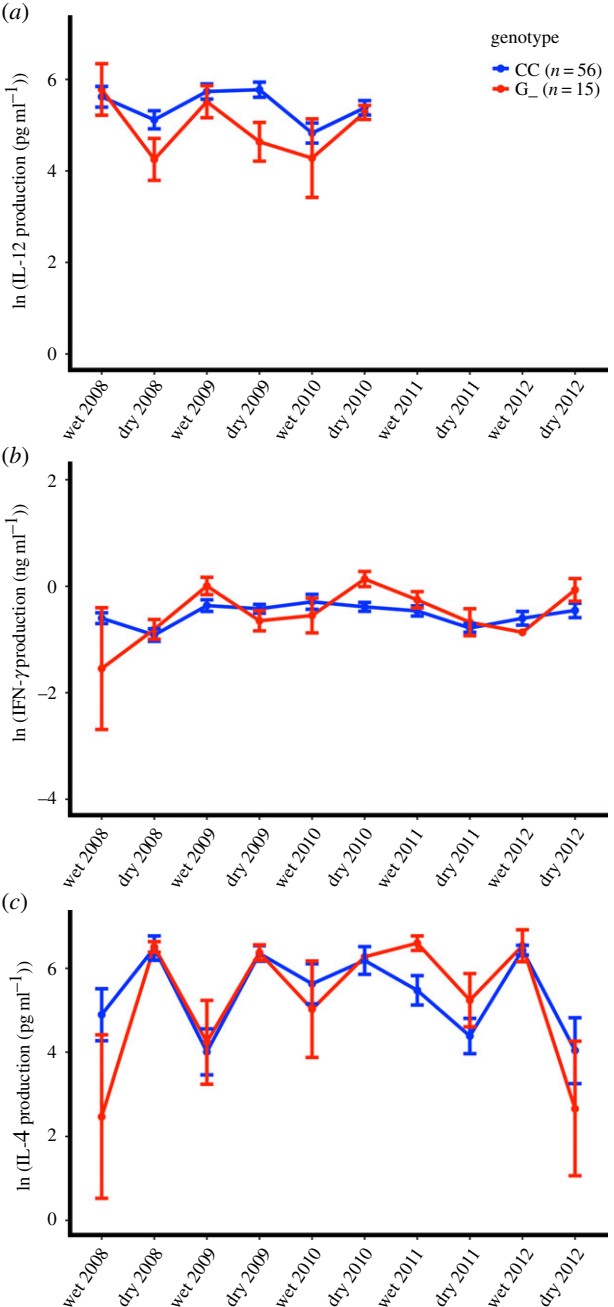

**Figure 3.** Cytokine production by SNP2253 genotype. (a) Animals heterozygous or homozygous for the 'G' risk allele produce 45% less IL-12 than CC animals following pokeweed mitogen stimulation of whole blood. By contrast, there is no detectable difference in (b) interferon gamma (IFNγ) or (c) interleukin 4 (IL-4) production relative to SNP2253 genotype in this population. (Online version in colour.)

congeners [5,6,10], demonstrating that simply giving disease a real-world environmental context can shift findings dramatically. Building upon this work, putative candidate genes for tolerance have been identified in wild rodents and tied to gene expression-based biomarkers, connecting animal-level disease to fine-scale gene expression patterns [11,12]. Multiscale studies can also uncover targets for treatment or intervention, by identifying genetic and immunologic mechanisms underlying variation in disease susceptibility while allowing for natural variation in hosts, microbes and environmental context. The findings presented here mirror immunological discoveries in controlled laboratory experiments in *M. tuberculosis*-infected mice [76–78]

and human cell lines [70,79,80], substantiating the important role of IL-12 and phagocyte activation in tuberculosis progression, even outside of the laboratory. In addition, multiscale studies that link genetic variation to disease dynamics and host fitness in wild systems provide an empirical foundation for models of host–pathogen coevolutionary dynamics, sharpening our ability to predict and understand long-term changes in host–pathogen interactions. Interestingly, bTB was only recently detected in this population of African buffalo in 1990 [36], representing a relatively new coevolutionary partner, which may explain why major alleles at these loci have not gone to fixation. Furthermore, bTB infection has been previously shown to increase mortality risk in this group of animals and animals infected earlier in life had lower reproductive rates [37,40]. Given the steep fitness costs of bTB infection, we would expect alleles associating with infection resistance to be under positive selection in this population. However, co-infection is common in this system [38,56,81], and fitness costs imposed by other pathogens may alter coevolutionary trajectories, resulting in the maintenance of variation at these bTB infection resistance loci.

Mycobacterial pathogens have evolved many mechanisms that directly modify host immune function, including the reduction of antigen presentation and activation in phagocytes and the active scavenging of reactive oxygen species (ROS) to avoid degradation [26]. Highly virulent strains of *M. tuberculosis* often escape the phagosome (where pathogens are gathered and degraded), prevent host cell apoptosis, and replicate in the cytosolic space, effectively avoiding a humoral immune response [29,31,32]. The balance between host immune recognition and mycobacterial evasion determines disease outcome at the animal level, leading to successful immune clearance or localized tissue necrosis and morbidity [30]. Phagocytes undergo activation through pathogen recognition, or by cytokine stimulation from $T_H1$ cells producing IFNγ [82]. Since pokeweed mitogens are used to test the efficacy of *T cell-dependent* immunity, we conclude that phagocytes classically activated by IFNγ are producing the IL-12 present in stimulated blood. Variation in IL-12 production suggests variation in phagocyte activation among SNP2253 genotypes, and may be a putative mechanism for variation in bTB resistance in this population. IL-12 has been repeatedly identified as an important cytokine in anti-mycobacterial immunity [33,34] and is highly upregulated locally in humans with active pulmonary tuberculosis [83–86]. Additionally, high variability in IL-12 expression has been demonstrated across murine families of varying *M. tuberculosis* susceptibility [83], mirroring the reduced IL-12 production observed here in susceptible buffalo. By contrast, we observed no difference in IFNγ or IL-4 production among SNP2253 genotypes. Thus, we conclude that the activation of $T_H1$ and $T_H2$ T-cells is not directly affected by variation at this locus.

Within activated macrophages, peroxisomes contribute to acidification of the phagosome through the production of ROS, directly affecting the cell's ability to degrade pathogens [75,87]. PEX14 is an integral peroxisomal protein [88,89], playing a critical role in both peroxisome formation and degradation, processes paramount to the ability to control and modify cellular ROS levels [90]. In plants, peroxisomal ROS production has been directly linked to inducible pre-invasion resistance mechanisms [91]. In animals, peroxisomes can modify cellular ROS concentrations leading to

macrophage apoptosis and increased local inflammation [87]. Interestingly, another gene near SNP2253, DNA fragmentation factor subunit alpha (*DFFA*; 35 kb upstream of SNP2253), plays an important role in the cellular machinery orchestrating apoptosis in response to infection [92]. Variation in these genes near SNP2253 may contribute to underlying variation in phagocyte-mediated pathogen degradation, apoptosis due to infection and the observed IL-12 inflammatory phenotype.

Free-ranging African buffalo and bTB provide a model system for understanding the ecological and evolutionary dynamics of mycobacterial pathogens that continue to plague humans, as well as wild and domestic animals. Here, we have identified putative mechanisms of bTB resistance that substantiate and build upon previous laboratory findings by exploring tuberculosis resistance within a natural population. We hope that future approaches in natural systems will complement advances from laboratory-based studies in understanding and outwitting one of humankind's most enduring scourges.

Ethics. Animals were kept under observation until fully recovered and all immobilizations were conducted by a veterinarian according to the South African National Parks Standard Operating Procedures for the Capture, Transportation and Maintenance in Holding Facilities of Wildlife. All animal work for this study was approved by the Institutional Animal Care and Use Committee (IACUC) at both Oregon State University (ACUP #3267) and the University of Georgia (UGA No. A201010Â¬190-A1), which follow the 8th edition of the Guide for the Care and Use of Laboratory Animals, the Guide for the Care and Use of Agricultural Animals in Research and Teaching and the European Convention for the Protection of Vertebrate Animals Used for Experimental and Other Scientific Purposes.

Data accessibility. All SNP, cytokine, infection and relevant covariate data are available on Dryad: doi:10.5061/dryad.3c45k28.

Authors' contributions. H.F.T. performed all molecular work, conducted all analyses and wrote this manuscript. V.O.E. and A.E.J. conceived the ideas for the main bTB project from which these data originated, designed the experiment, oversaw data collection and obtained funding for the work. E.G.H., N.l.R. and P.D.v.H. provided the *Syncerus caffer* genome and advice on early analyses. All authors contributed critically to drafts and gave final approval for publication.

Competing interests. We declare we have no competing interests.

Funding. This research was supported by the National Science Foundation through an Ecology of Infectious Diseases award to V.O.E. and A.E.J. (EF-0723918, DEB-1102493/EF-0723928), and a Morris Animal Foundation Pilot Grant awarded to H.F.T. and A.E.J. (D15ZO-824). E.G.H. and P.D.v.H. acknowledge funding from South African MRC and NRF. Additional support came from the National Institutes of Health grant no. UG3 OD023389 and an American Association of University Women (AAUW) Fellowship awarded to H.F.T.

Acknowledgements. Buffalo captures and immune assays were conducted by Brianna Beechler, Sarah Budishak, Erin Gorsich, Paul Snyder, Robert Spaan, Johannie Spaan and Kristie Thompson. The authors thank Eli Meyer and Holland Elder for their training and guidance in RAD-sequencing techniques and bioinformatics, and Henri Combrink for his assistance with the InterProScan analyses.

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
