## [Reviewer comments · Proceedings of the Royal Society B: Biological Sciences]

Review History

RSPB-2019-0077.R0 (Original submission)

Review form: Reviewer 1

Recommendation

Reject – article is not of sufficient interest (we will consider a transfer to another journal)

Scientific importance: Is the manuscript an original and important contribution to its field?

Good

General interest: Is the paper of sufficient general interest?

Good

Quality of the paper: Is the overall quality of the paper suitable?

Good

Is the length of the paper justified?

Yes

Should the paper be seen by a specialist statistical reviewer?

No

Do you have any concerns about statistical analyses in this paper? If so, please specify them explicitly in your report.

No

It is a condition of publication that authors make their supporting data, code and materials available - either as supplementary material or hosted in an external repository. Please rate, if applicable, the supporting data on the following criteria.

Is it accessible?

Yes

Is it clear?

Yes

Is it adequate?

Yes

Do you have any ethical concerns with this paper?

No

Comments to the Author

In this study, the authors explore the genetic basis for resistance to bovine tuberculosis in a natural population of African buffalo. They identify two resistance loci, one of which is associated with variation in IL12 production, itself associated with phagocyte activation. Their results suggest distinct inflammatory phenotypes among buffalo, with consequences for risk of bTB infection in the wild. As the authors point out, studies to date have been limited to the laboratory, which is far removed from the reality faced by animals in the wild. As a result, little is known about the basis of different immunological strategies, such as resistance or tolerance, in the wild. This is an important area of research, given its potential application to the control and treatment of infectious disease in the wild. I applaud the authors for addressing this important knowledge gap in a complex natural system, and for applying a holistic approach to their study - weaving together evidence from genetics, immunology and infection data. However, I do have some concerns.

I would strongly suggest that the authors change the title to more clearly represent the findings of their study. The introduction lacks detail on the (very interesting) buffalo-bTB system, which is necessary to set the scene for the rest of the paper. For example, why is bTB important? The authors mention at the end of the MS, that it is "one of humankind's most enduring scourges" but provide no evidence to back this statement up. The manuscript also lacks methodological detail in place (please see more detailed comments below). Finally, the discussion is very mechanistic in places, and (in its current form) more suited for a specialised audience.

I have included some more detailed comments below:

76 & 186: Please be clear about the findings of your previous study. I believe you didn't actually detect heritable variation in this trait.

123: Please include the 95% confidence interval for this estimate of prevalence

126: Here and throughout "INF" should be changed to "IFN"

124 - 131: These references are all very general/not buffalo-specific. Are there any more specific

studies that could be referenced here? If not, then it should be made clear that these statement are based on knowledge from other systems.

138: Please include more explanation re. stimulation with pokeweed i.e. testing potential of the immune system to respond to challenge.

140 – 145: These details should be included in a separate statistical section

143: Why were some cytokine measures missing?

173 – 175: This is unnecessary detail, I would suggest removing

159 – 62: This statement is too vague. Please include details.

147 – 182: The majority of the genotyping methods are currently copied and pasted from the authors' previous paper (ref no. 30). I would suggest referring to this paper in places, to shorten this section/avoid repetition.

261 – 263: These details are repeated (already mention on line 210), I would suggest removing one of these instances.

197: Please include a sentence somewhere explicitly explaining why anti-helminthic treatment (though not the focus of this study) was accounted for in your models.

198: Were potential temporal effects accounted for in this model?

198 & 225 – 227: Were fixed effects checked for collinearity? Please include details in MS.

201: This is too vague. Please include more details e.g. what percentage of variation was explained by these 5 components?

213: Please provide a brief description of how InterProScan works.

226: Please include a justification for the inclusion of these interaction terms

232 – 235: “Shapiro Wilk” not “Shapiro Wilks”. This statement is also unclear. Were models which violated the assumption of normality discarded? If so, how were models adjusted to ensure that residuals met this assumption? Were residuals also checked to ensure they were homoscedastic? Please include more detail here.

233: This statement re. model selection is also unclear. Elsewhere, you refer to “AIC-based selection”, please clarify.

284 – 287: You have not cited or discussed two relevant papers from a wild rodent system which show polymorphism at a specific gene associated with differences in the Th2 immune response and tolerance to macroparasites (Jackson et al. 2014 PLoS Biol 12; Wanelik et al. 2018 Mol Ecol 27).

Review form: Reviewer 2

Recommendation

Accept with minor revision (please list in comments)

Scientific importance: Is the manuscript an original and important contribution to its field?

Excellent

General interest: Is the paper of sufficient general interest?

Excellent

Quality of the paper: Is the overall quality of the paper suitable?

Excellent

Is the length of the paper justified?

Yes

Should the paper be seen by a specialist statistical reviewer?

No

Do you have any concerns about statistical analyses in this paper? If so, please specify them explicitly in your report.

No

It is a condition of publication that authors make their supporting data, code and materials available - either as supplementary material or hosted in an external repository. Please rate, if applicable, the supporting data on the following criteria.

Is it accessible?

Yes

Is it clear?

Yes

Is it adequate?

Yes

Do you have any ethical concerns with this paper?

No

Comments to the Author

In this manuscript, the authors study risk of bovine tuberculosis (bTB) infection in populations of wild African buffalo, and identify two genetic loci in those populations that additively increase risk of bTB infection. The high risk allele at one of those loci correlates with reduced IL-12 production, suggesting lower macrophage activation. This fascinating study links genotypic variation in wild host populations to immune function (i.e., a putative mechanism) and risk of infection with a globally important disease. The manuscript is extremely well written, and I have only very minor comments.

Minor comments

L65: Typo: change "evidences" to "evidence"

L110-145: I found the methods presentation in the "bTB Testing and Cytokine Stimulations" section to be a little confusing. Specifically:

1. Were all of the blood samples (including the sample for bTB testing) from peripheral blood from the jugular vein? If not, from where on the buffalo did the blood for bTB testing come from?
2. To help non-immunologists (like me...) understand the methods, I think it would be good if you used the word "cytokine" at least once in the paragraph in lines 124-131. That paragraph sets the stage for what variables you were interested in measuring, and why (which is great!), but doesn't refer to any of the immune components as "cytokines". Then, the subsequent paragraph (lines 132-145) tells exactly how the interferon gamma, IL-12, and IL-4 were measured, but only refers to those components as "cytokines." I am embarrassed to say that I had to Google the word "cytokine" just to confirm that the interferon gamma, interleukin 12, and interleukin 4 were indeed what lines 132-145 were talking about.

L117-122: Just to be clear, you obtained time series of 2-9 bTB tests for each animal, but only animals with time series of 4-9 bTB tests were included in the analysis (i.e., animals with fewer than 4 test results were excluded)? If so, I feel this could be stated a little more clearly.

L159: Typo: Alf1 is written Alfl here.

L252: I think this sentence would be improved by changing “though” to “and” (“These two loci had an additive effect on bTB conversion risk, *and* no interactive effects were detected.”

Decision letter (RSPB-2019-0077.R0)

13-Mar-2019

Dear Dr Tavalire:

I am writing to inform you that your manuscript RSPB-2019-0077 entitled "Variation in tuberculosis resistance persists across scales in a wild mammalian host" has, in its current form, been rejected for publication in Proceedings B.

This action has been taken on the advice of referees, who have recommended that substantial revisions are necessary. With this in mind we would be happy to consider a resubmission, provided the comments of the referees are fully addressed. However please note that this is not a provisional acceptance.

Sincerely,
Proceedings B
<mailto:proceedingsb@royalsociety.org>

Associate Editor
Board Member: 1
Comments to Author:

In this article, the authors seek to gain insights into the genetic and immunological basis of variation in disease susceptibility to the bacterial pathogen *Mycobacterium bovis* in wild

populations of the African buffalo. This is an interesting and well written article. I think linking the different levels of organisation (genetics, immunology and epidemiology) in this way is very valuable and has rarely been done before in wild species. However, both reviewers have identified a number of issues that need to be addressed. Most importantly, the authors need to make a stronger effort throughout the manuscript to make the ms more accessible to the non-specialist reader. In the Introduction, it would also be important to provide much more background on bTB. Although the broad host range of *M. bovis* is briefly mentioned, the importance of this pathogen in agriculture, ecology and public health does not come across at all. Also, more details on the statistical methods need to be provided, as pointed out by reviewer 1.

Some more specific comments:

- I agree with reviewer 1 that the title is a too vague. To me, the word "persists" implies that variation in resistance is maintained over some period of time whereas in fact there is resistance, genetic, and immunological variation that is correlated. It's also not immediately clear what "across scales" refers to.

- 1.203: I assume that q is the false discovery rate? If so, this should be explicitly stated, and some justification should be given for the two values that were chosen.

- Figure 2: I might have missed something here but I was a bit surprised that the curves only start to drop off at around 4 years of age. Do younger buffalos not get infected at all?

Reviewer(s)' Comments to Author:

Referee: 1

Comments to the Author(s)

In this study, the authors explore the genetic basis for resistance to bovine tuberculosis in a natural population of African buffalo. They identify two resistance loci, one of which is associated with variation in IL12 production, itself associated with phagocyte activation. Their results suggest distinct inflammatory phenotypes among buffalo, with consequences for risk of bTB infection in the wild. As the authors point out, studies to date have been limited to the laboratory, which is far removed from the reality faced by animals in the wild. As a result, little is known about the basis of different immunological strategies, such as resistance or tolerance, in the wild. This is an important area of research, given its potential application to the control and treatment of infectious disease in the wild. I applaud the authors for addressing this important knowledge gap in a complex natural system, and for applying a holistic approach to their study - weaving together evidence from genetics, immunology and infection data. However, I do have some concerns.

I would strongly suggest that the authors change the title to more clearly represent the findings of their study. The introduction lacks detail on the (very interesting) buffalo-bTB system, which is necessary to set the scene for the rest of the paper. For example, why is bTB important? The authors mention at the end of the MS, that it is "one of humankind's most enduring scourges" but provide no evidence to back this statement up. The manuscript also lacks methodological detail in place (please see more detailed comments below). Finally, the discussion is very mechanistic in places, and (in its current form) more suited for a specialised audience.

I have included some more detailed comments below:

- 76 & 186: Please be clear about the findings of your previous study. I believe you didn't actually detect heritable variation in this trait.
- 123: Please include the 95% confidence interval for this estimate of prevalence
- 126: Here and throughout "INF" should be changed to "IFN"
- 124 - 131: These references are all very general/not buffalo-specific. Are there any more specific studies that could be referenced here? If not, then it should be made clear that these statement are based on knowledge from other systems.
- 138: Please include more explanation re. stimulation with pokeweed i.e. testing potential of the immune system to respond to challenge.
- 140 - 145: These details should be included in a separate statistical section
- 143: Why were some cytokine measures missing?
- 173 - 175: This is unnecessary detail, I would suggest removing
- 159 - 62: This statement is too vague. Please include details.
- 147 - 182: The majority of the genotyping methods are currently copied and pasted from the authors' previous paper (ref no. 30). I would suggest referring to this paper in places, to shorten this section/avoid repetition.
- 261 - 263: These details are repeated (already mention on line 210), I would suggest removing one of these instances.
- 197: Please include a sentence somewhere explicitly explaining why anti-helminthic treatment (though not the focus of this study) was accounted for in your models.
- 198: Were potential temporal effects accounted for in this model?
- 198 & 225 - 227: Were fixed effects checked for collinearity? Please include details in MS.
- 201: This is too vague. Please include more details e.g. what percentage of variation was explained by these 5 components?
- 213: Please provide a brief description of how InterProScan works.
- 226: Please include a justification for the inclusion of these interaction terms
- 232 - 235: "Shapiro Wilk" not "Shapiro Wilks". This statement is also unclear. Were models which violated the assumption of normality discarded? If so, how were models adjusted to ensure that residuals met this assumption? Were residuals also checked to ensure they were homoscedastic? Please include more detail here.
- 233: This statement re. model selection is also unclear. Elsewhere, you refer to "AIC-based selection", please clarify.
- 284 - 287: You have not cited or discussed two relevant papers from a wild rodent system which show polymorphism at a specific gene associated with differences in the Th2 immune response and tolerance to macroparasites (Jackson et al. 2014 PLoS Biol 12; Wanelik et al. 2018 Mol Ecol 27).

Referee: 2

Comments to the Author(s)

In this manuscript, the authors study risk of bovine tuberculosis (bTB) infection in populations of wild African buffalo, and identify two genetic loci in those populations that additively increase risk of bTB infection. The high risk allele at one of those loci correlates with reduced IL-12 production, suggesting lower macrophage activation. This fascinating study links genotypic variation in wild host populations to immune function (i.e., a putative mechanism) and risk of infection with a globally important disease. The manuscript is extremely well written, and I have only very minor comments.

Minor comments

L65: Typo: change "evidences" to "evidence"

L110-145: I found the methods presentation in the “bTB Testing and Cytokine Stimulations” section to be a little confusing. Specifically:

1. Were all of the blood samples (including the sample for bTB testing) from peripheral blood from the jugular vein? If not, from where on the buffalo did the blood for bTB testing come from?
2. To help non-immunologists (like me...) understand the methods, I think it would be good if you used the word “cytokine” at least once in the paragraph in lines 124-131. That paragraph sets the stage for what variables you were interested in measuring, and why (which is great!), but doesn’t refer to any of the immune components as “cytokines”. Then, the subsequent paragraph (lines 132-145) tells exactly how the interferon gamma, IL-12, and IL-4 were measured, but only refers to those components as “cytokines.” I am embarrassed to say that I had to Google the word “cytokine” just to confirm that the interferon gamma, interleukin 12, and interleukin 4 were indeed what lines 132-145 were talking about.

L117-122: Just to be clear, you obtained time series of 2-9 bTB tests for each animal, but only animals with time series of 4-9 bTB tests were included in the analysis (i.e., animals with fewer than 4 test results were excluded)? If so, I feel this could be stated a little more clearly.

L159: Typo: Alf1 is written Alfl here.

L252: I think this sentence would be improved by changing “though” to “and” (“These two loci had an additive effect on bTB conversion risk, *and* no interactive effects were detected.”

Author's Response to Decision Letter for (RSPB-2019-0077.R0)

See Appendix A.

RSPB-2019-0914.R0

Review form: Reviewer 1

Recommendation

Accept with minor revision (please list in comments)

Scientific importance: Is the manuscript an original and important contribution to its field?

Good

General interest: Is the paper of sufficient general interest?

Excellent

Quality of the paper: Is the overall quality of the paper suitable?

Excellent

Is the length of the paper justified?

Yes

Should the paper be seen by a specialist statistical reviewer?

No

Do you have any concerns about statistical analyses in this paper? If so, please specify them explicitly in your report.

No

It is a condition of publication that authors make their supporting data, code and materials available - either as supplementary material or hosted in an external repository. Please rate, if applicable, the supporting data on the following criteria.

Is it accessible?

Yes

Is it clear?

Yes

Is it adequate?

Yes

Do you have any ethical concerns with this paper?

No

Comments to the Author

I am very happy with the authors' thorough responses to my concerns, and believe the paper is much improved. I have two outstanding comments:

(1) My original comment (line 198) referred to the possibility of including a year effect in the Cox regression model i.e. which year an individual was sampled in. Could this affect conversion risk?

(2) I am still slightly concerned about the authors' representation of the novelty of this study. I would suggest changing lines 330-331 to read: "This is one of very few studies taking a multi-level approach, connecting disease phenotypes across scales, from an individual-level disease trait to patterns in immune reactivity in the wild."

Review form: Reviewer 3

Recommendation

Accept with minor revision (please list in comments)

Scientific importance: Is the manuscript an original and important contribution to its field?

Excellent

General interest: Is the paper of sufficient general interest?

Good

Quality of the paper: Is the overall quality of the paper suitable?

Good

Is the length of the paper justified?

Yes

Should the paper be seen by a specialist statistical reviewer?

Yes

Do you have any concerns about statistical analyses in this paper? If so, please specify them explicitly in your report.

No

It is a condition of publication that authors make their supporting data, code and materials available - either as supplementary material or hosted in an external repository. Please rate, if applicable, the supporting data on the following criteria.

Is it accessible?

Yes

Is it clear?

Yes

Is it adequate?

Yes

Do you have any ethical concerns with this paper?

No

Comments to the Author

The authors present the results of an innovative study, in which they link findings from the genome- to the population-level to identify genes that encode for elevated risk of positive infection with *Mycobacterium tuberculosis* in a wild African buffalo study system. The authors convincingly demonstrate that buffalo possessing one of two single nucleotide polymorphisms (SNPs) identified in their analysis are statistically more likely to contract Tb at an earlier age than those lacking the polymorphism. These genes appear to be additive, meaning that those individuals positive for both SNPs are at the greatest infection hazard of all. The authors link these genotypes to immune phenotype by demonstrating that individuals positive for one of their self-identified "risk alleles" demonstrate reduced production of IL-12, an important cytokine in phagocyte activation and anti-Tb defenses, across their lifetimes, suggesting a mechanistic basis for reduced resistance to infection. The authors further show that these genome variations are not associated with any variation in lifetime production of IFN-gamma or IL-4, indicating no SNP-associated variation in Th1 or Th2-primed immunity. Rather, these genetic anomalies appear to be linked to IL-12 and anti-Tb defenses specifically.

In general, I find this to be a compelling and fascinating study that is noble in its efforts to bridge between within-host immunological mechanisms and population-level ecological patterns. I have a few concerns with a lack of precision in some of the language, especially in the introduction, concerning definitions of the terms "resistance" and "tolerance" as well as "disease." In particular, the authors highlight high population-level maintenance of Tb in the African buffalo system (lines 88-97) but then emphasize resistance alleles that would limit infection rates for these host. Please resolve these inconsistencies or discuss how variation in individual resistance might contribute to high population-level prevalence (perhaps by restricting infection to later age individuals and reducing lifelong mortality effects). Re: use of the term "disease," I would swap this for "infection" when possible to avoid needing to demonstrate some sort of pathological manifestation in the host. The authors present no information on the morbidity or mortality

effects of Tb infection for buffalo (though see comments on Discussion for how some of this might be helpful to understand evolutionary pressures in this section).

Additionally, I think the intro could be better organized. As outlined below, I suggest restructuring the paragraphs into the following order: (a) intro to the rarity of cross-scale studies, (b) intro to the global burden of Mycobacterium across species, including humans and cattle, (c) intro to Tb's methods of immune avoidance, (d) highlight how buffalo are effective at resisting infection nonetheless, suggesting some heritable resistance traits (but be careful to follow precise definitions of resistance and tolerance), and (e) use this to motivate the imperative in understanding resistance mechanisms against this pathogen so as to understand infection risks in other hosts.

Specific comments:

Line numbers refer to the clean, corrected draft in the absence of track changes.

Abstract:

Line 4: Suggest adding commas around "along with environmental factors" to eliminate confusion from sentence structure

Line 7: Suggestion to drop the term "transmission" as it is not often observable in the wild but usually inferred via model fitting – and it is not actually observed here. In this study, we witness evidence of infection conversion but the actual transmission event is not demonstrated (and is, to my knowledge, quite poorly understood). Given the authors' desire to highlight something that is "uniquely observable in natural populations", I suggest the following rewording: "However, genetics and immunology are typically studied in model systems, whereas population-level patterns of infection status and susceptibility are uniquely observable in nature."

Also, I'm not a huge fan of the term "disease" as it carries so many implications about effects on host health.

Line 11: Keep to present tense to be consistent with the rest of the abstract

Intro:

Again, reiterating my objection to the term "disease" or at least a request that it be crisply defined...

Line 23: you need to define what you mean by 'tolerant' since it is a loaded term. I'm assuming this is the ecological definition, meaning that hosts carry high pathogen burdens but that their health is not impacted?

Line 48: Would be good to define "effector cell" here, since the paper claims to be targeting a wide range of sub-disciplines for readership

Line 52: How are pathogen-recognition mechanisms integral to pathogen survival? Please clarify what you mean here.

Line 68: Should be two sentences here. Also, the motivation for the study is a bit weak – why do you look for resistance mechanisms if limited evidence of this exists? Suggestion to restructure the whole intro section as follows: (a) intro cross-scale studies as done here, (b) intro to the burden of Mycobacterium across species and global human population, (c) highlight its crafty methods of immune avoidance, (d) but highlight how buffalo are effective at tolerating it nonetheless, and (e)

use this to motivate the imperative in understanding resistance mechanisms against this pathogen so as to alleviate disease burden in other hosts.

Also, be careful with terms – you talk a lot about “resistance loci” but then say that buffalo sustain high levels of infection...It is certainly possible for both resistance and tolerance to be at play, but you need to make sure you use your terms correctly and carefully.

Methods

In general, the methods are described thoroughly, succinctly, and clearly. As I am not a population geneticist myself, I cannot assess SNP Genotyping and Filtering methods described in this paper and suggest to the editor that at least one reviewer in this field should comment on the manuscript.

Line 250: spp. should not be italicized here.

Results

Line 313 and Figure 3: Figure 1 suggests that animals possessing both “G_” and “T_” risk alleles are at a compounded risk of obtaining positive Tb infection, over and above those heterozygous for each. Can you include separate curves for each of these genotypes in Fig 3 as well, so that we can visualize allele-specific differences in effects on immune function? From your mixed effects model (Table 3), it appears that only the “G_” allele is significantly associated with reduced IL-12 production. In discussion of these results in the text, you do not address the absence of relationship for the “T_” allele with cytokine production patterns. Please report these results and include a line or two in the Discussion speculating on the mechanism of this gene’s effects on Tb susceptibility.

Line 316: I’m not sure how Fig 3 shows that IL-12 is not affected by Tb status. Suggestion to remove this reference and include only the ref to Table 3 (which demonstrates this lack of association clearly).

Discussion

In general, discussion is clear and concise. As mentioned above, I’d like to see some discussion of the interactive effects of the two SNPs identified on infection risk and the absence of effect for the second allele on cytokine production. Are there other immune factors that the authors did not assay that could underly the mechanism of this genotype’s effects on infection risk?

Additionally, some discussion of the evolutionary underpinnings and maintenance of the identified SNPs would be helpful. How are these SNPs maintained? What is the mortality hazard imposed by Tb infection on African buffalo? The authors mention in the introduction that high infection rates are common throughout wild populations but do not indicate whether those positive for infection are also more likely to die early. It would be good to have at least some sense of the evolutionary selective environment in which these SNPs have arisen and been maintained.

Decision letter (RSPB-2019-0914.R0)

28-May-2019

Dear Dr Tavalire:

Your manuscript has now been peer reviewed and the reviews have been assessed by an

Associate Editor. The reviewers' comments (not including confidential comments to the Editor) and the comments from the Associate Editor are included at the end of this email for your reference. As you will see, the reviewers and the Editors have raised some concerns with your manuscript and we would like to invite you to revise your manuscript to address them.

Research ethics:

Use of animals and field studies:

If you wish to submit your data to Dryad (<http://datadryad.org/>) and have not already done so

you can submit your data via this link

Please submit a copy of your revised paper within three weeks. If we do not hear from you within this time your manuscript will be rejected. If you are unable to meet this deadline please let us know as soon as possible, as we may be able to grant a short extension.

Best wishes,
Proceedings B
mailto: proceedingsb@royalsociety.org

Associate Editor Board Member

Comments to Author:

The authors have done an excellent job at addressing the reviewers' comments on their previous submission. The paper has now also been reviewed by a 3rd reviewer who is also very positive. There are some outstanding comments from both reviewers 1 and 3 that would need to be addressed but most of these comments relate to improving the clarity of presentation and should therefore be relatively straightforward to address.

I agree with reviewer 3's suggestion of further improving the introduction. As it stands, statements on general immunological principles, immunology of Mycobacterium and epidemiological statements on Mycobacterium are all intermingled within paragraphs. It would be good to better separate those statements and the structure suggested by Reviewer 3 seems suitable to achieve this.

Another, very minor comment about the Introduction: I found the statement in lines 78-79 that *M. bovis* makes up a larger fraction of human TB than previously estimated very interesting but felt it was rather vague. It would be good if the authors could provide some actual numbers here. (It

makes a big difference whether previous estimates were 0% and now they are 5% or whether they were previously 0.001% and now they are 0.0015%!

Reviewer(s)' Comments to Author:

Referee: 1

Comments to the Author(s).

I am very happy with the authors' thorough responses to my concerns, and believe the paper is much improved. I have two outstanding comments:

(1) My original comment (line 198) referred to the possibility of including a year effect in the Cox regression model i.e. which year an individual was sampled in. Could this affect conversion risk?

(2) I am still slightly concerned about the authors' representation of the novelty of this study. I would suggest changing lines 330-331 to read: "This is one of very few studies taking a multi-level approach, connecting disease phenotypes across scales, from an individual-level disease trait to patterns in immune reactivity in the wild."

Referee: 3

Comments to the Author(s).

The authors present the results of an innovative study, in which they link findings from the genome- to the population-level to identify genes that encode for elevated risk of positive infection with *Mycobacterium tuberculosis* in a wild African buffalo study system. The authors convincingly demonstrate that buffalo possessing one of two single nucleotide polymorphisms (SNPs) identified in their analysis are statistically more likely to contract Tb at an earlier age than those lacking the polymorphism. These genes appear to be additive, meaning that those individuals positive for both SNPs are at the greatest infection hazard of all. The authors link these genotypes to immune phenotype by demonstrating that individuals positive for one of their self-identified "risk alleles" demonstrate reduced production of IL-12, an important cytokine in phagocyte activation and anti-Tb defenses, across their lifetimes, suggesting a mechanistic basis for reduced resistance to infection. The authors further show that these genome variations are not associated with any variation in lifetime production of IFN-gamma or IL-4, indicating no SNP-associated variation in Th1 or Th2-primed immunity. Rather, these genetic anomalies appear to be linked to IL-12 and anti-Tb defenses specifically.

In general, I find this to be a compelling and fascinating study that is noble in its efforts to bridge between within-host immunological mechanisms and population-level ecological patterns. I have a few concerns with a lack of precision in some of the language, especially in the introduction, concerning definitions of the terms "resistance" and "tolerance" as well as "disease." In particular, the authors highlight high population-level maintenance of Tb in the African buffalo system (lines 88-97) but then emphasize resistance alleles that would limit infection rates for these host. Please resolve these inconsistencies or discuss how variation in individual resistance might contribute to high population-level prevalence (perhaps by restricting infection to later age individuals and reducing lifelong mortality effects). Re: use of the term "disease," I would swap this for "infection" when possible to avoid needing to demonstrate some sort of pathological manifestation in the host. The authors present no information on the morbidity or mortality effects of Tb infection for buffalo (though see comments on Discussion for how some of this might be helpful to understand evolutionary pressures in this section).

Additionally, I think the intro could be better organized. As outlined below, I suggest restructuring the paragraphs into the following order: (a) intro to the rarity of cross-scale studies,

(b) intro the global burden of Mycobacterium across species, including humans and cattle, (c) intro to Tb's methods of immune avoidance, (d) highlight how buffalo are effective at resisting infection nonetheless, suggesting some heritable resistance traits (but be careful to follow precise definitions of resistance and tolerance), and (e) use this to motivate the imperative in understanding resistance mechanisms against this pathogen so as to understand infection risks in other hosts.

Specific comments:

Line numbers refer to the clean, corrected draft in the absence of track changes.

Abstract:

Line 4: Suggest adding commas around “along with environmental factors” to eliminate confusion from sentence structure

Line 7: Suggestion to drop the term “transmission” as it is not often observable in the wild but usually inferred via model fitting – and it is not actually observed here. In this study, we witness evidence of infection conversion but the actual transmission event is not demonstrated (and is, to my knowledge, quite poorly understood). Given the authors’ desire to highlight something that is “uniquely observable in natural populations”, I suggest the following rewording: “However, genetics and immunology are typically studied in model systems, whereas population-level patterns of infection status and susceptibility are uniquely observable in nature.”

Also, I’m not a huge fan of the term “disease” as it carries so many implications about effects on host health.

Line 11: Keep to present tense to be consistent with the rest of the abstract

Intro:

Again, reiterating my objection to the term “disease” or at least a request that it be crisply defined...

Line 23: you need to define what you mean by ‘tolerant’ since it is a loaded term. I’m assuming this is the ecological definition, meaning that hosts carry high pathogen burdens but that their health is not impacted?

Line 48: Would be good to define “effector cell” here, since the paper claims to be targeting a wide range of sub-disciplines for readership

Line 52: How are pathogen-recognition mechanisms integral to pathogen survival? Please clarify what you mean here.

Line 68: Should be two sentences here. Also, the motivation for the study is a bit weak – why do you look for resistance mechanisms if limited evidence of this exists? Suggestion to restructure the whole intro section as follows: (a) intro cross-scale studies as done here, (b) intro the burden of Mycobacterium across species and global human population, (c) highlight its crafty methods of immune avoidance, (d) but highlight how buffalo are effective at tolerating it nonetheless, and (e) use this to motivate the imperative in understanding resistance mechanisms against this pathogen so as to alleviate disease burden in other hosts.

Also, be careful with terms – you talk a lot about “resistance loci” but then say that buffalo sustain high levels of infection...It is certainly possible for both resistance and tolerance to be at play, but you need to make sure you use your terms correctly and carefully.

Methods

In general, the methods are described thoroughly, succinctly, and clearly. As I am not a population geneticist myself, I cannot assess SNP Genotyping and Filtering methods described in this paper and suggest to the editor that at least one reviewer in this field should comment on the manuscript.

Line 250: spp. should not be italicized here.

Results

Line 313 and Figure 3: Figure 1 suggests that animals possessing both “G_” and “T_” risk alleles are at a compounded risk of obtaining positive Tb infection, over and above those heterozygous for each. Can you include separate curves for each of these genotypes in Fig 3 as well, so that we can visualize allele-specific differences in effects on immune function? From your mixed effects model (Table 3), it appears that only the “G_” allele is significantly associated with reduced IL-12 production. In discussion of these results in the text, you do not address the absence of relationship for the “T_” allele with cytokine production patterns. Please report these results and include a line or two in the Discussion speculating on the mechanism of this gene’s effects on Tb susceptibility.

Line 316: I’m not sure how Fig 3 shows that IL-12 is not affected by Tb status. Suggestion to remove this reference and include only the ref to Table 3 (which demonstrates this lack of association clearly).

Discussion

In general, discussion is clear and concise. As mentioned above, I’d like to see some discussion of the interactive effects of the two SNPs identified on infection risk and the absence of effect for the second allele on cytokine production. Are there other immune factors that the authors did not assay that could underly the mechanism of this genotype’s effects on infection risk?

Additionally, some discussion of the evolutionary underpinnings and maintenance of the identified SNPs would be helpful. How are these SNPs maintained? What is the mortality hazard imposed by Tb infection on African buffalo? The authors mention in the introduction that high infection rates are common throughout wild populations but do not indicate whether those positive for infection are also more likely to die early. It would be good to have at least some sense of the evolutionary selective environment in which these SNPs have arisen and been maintained.

Author's Response to Decision Letter for (RSPB-2019-0914.R0)

See Appendix B.

Decision letter (RSPB-2019-0914.R1)

17-Jun-2019

Dear Dr Tavalire

I am pleased to inform you that your Review manuscript RSPB-2019-0914.R1 entitled "Risk alleles for tuberculosis infection associate with reduced immune reactivity in a wild mammalian host" has been accepted for publication in Proceedings B.

The referee(s) do not recommend any further changes. Therefore, please proof-read your manuscript carefully and upload your final files for publication. Because the schedule for publication is very tight, it is a condition of publication that you submit the revised version of your manuscript within 7 days. If you do not think you will be able to meet this date please let me know immediately.

To upload your manuscript, log into <http://mc.manuscriptcentral.com/prsb> and enter your Author Centre, where you will find your manuscript title listed under "Manuscripts with Decisions." Under "Actions," click on "Create a Revision." Your manuscript number has been appended to denote a revision.

You will be unable to make your revisions on the originally submitted version of the manuscript. Instead, upload a new version through your Author Centre.

1) A text file of the manuscript (doc, txt, rtf or tex), including the references, tables (including captions) and figure captions. Please remove any tracked changes from the text before submission. PDF files are not an accepted format for the "Main Document".

2) A separate electronic file of each figure (tiff, EPS or print-quality PDF preferred). The format should be produced directly from original creation package, or original software format. Please note that PowerPoint files are not accepted.

3) Electronic supplementary material: this should be contained in a separate file from the main text and the file name should contain the author's name and journal name, e.g. `authorname_procb_ESM_figures.pdf`

All supplementary materials accompanying an accepted article will be treated as in their final form. They will be published alongside the paper on the journal website and posted on the online figshare repository. Files on figshare will be made available approximately one week before the accompanying article so that the supplementary material can be attributed a unique DOI. Please see: <https://royalsociety.org/journals/authors/author-guidelines/>

4) Data-Sharing and data citation

It is a condition of publication that data supporting your paper are made available. Data should be made available either in the electronic supplementary material or through an appropriate repository. Details of how to access data should be included in your paper. Please see <https://royalsociety.org/journals/ethics-policies/data-sharing-mining/> for more details.

<http://datadryad.org/submit?journalID=RSPB&manu=RSPB-2019-0914.R1> which will take you to your unique entry in the Dryad repository.

Once again, thank you for submitting your manuscript to Proceedings B and I look forward to receiving your final version. If you have any questions at all, please do not hesitate to get in touch.

Sincerely,

Professor Gary Carvalho
Editor, Proceedings B
<mailto:proceedingsb@royalsociety.org>

Associate Editor

Comments to Author:

The authors have addressed all remaining issues in a thorough manner. In particular, I think the Background is now very clear and easy to follow. I only have a few more minor comments:

- l. 344 & 350: I may have missed something here but why "genes"? Wasn't PEX14 the only identified candidate gene in the vicinity of the two SNPs?
- l. 351: I think it's not the locus that increases risk of bTB but an allele in this region that appears to be associated with increased risk.
- l. 368: I don't think this second "foundational" is necessary.

Decision letter (RSPB-2019-0914.R2)

24-Jun-2019

Dear Dr Tavalire

I am pleased to inform you that your manuscript entitled "Risk alleles for tuberculosis infection associate with reduced immune reactivity in a wild mammalian host" has been accepted for publication in Proceedings B.

Open Access

Paper charges

Sincerely,

Appendix A

Response to Referees

We thank the associate editor and two anonymous reviewers for their constructive comments on this manuscript. Resulting edits have greatly strengthened the manuscript and we appreciate the attention to detail. Please see broad and specific responses to each comment below in italics.

Associate Editor

Board Member: 1

Comments to Author:

In this article, the authors seek to gain insights into the genetic and immunological basis of variation in disease susceptibility to the bacterial pathogen *Mycobacterium bovis* in wild populations of the African buffalo. This is an interesting and well written article. I think linking the different levels of organisation (genetics, immunology and epidemiology) in this way is very valuable and has rarely been done before in wild species. However, both reviewers have identified a number of issues that need to be addressed. Most importantly, the authors need to make a stronger effort throughout the manuscript to make the ms more accessible to the non-specialist reader. In the Introduction, it would also be important to provide much more background on bTB. Although the broad host range of *M. bovis* is briefly mentioned, the importance of this pathogen in agriculture, ecology and public health does not come across at all. Also, more details on the statistical methods need to be provided, as pointed out by reviewer 1.

- *We feel that we have sufficiently addressed the broad and specific comments of all reviewers pertaining to accessibility by reducing much of the immunology jargon in the introduction and discussion and using language that conveys higher-level immune processes instead of naming specific receptors, proteins, etc, when more precise language was not needed (e.g., lines 49-61, 443-447, 461-468, and 470-501). However, we have retained details we feel are integral to connecting the putative candidate genes to the IL-12 story. We are happy to revisit these edits if needed. Additionally, we have added background information to the introduction about bTB and tuberculosis in general to more appropriately set the stage for this widespread, important pathogen (lines 86-98). We have also added extensive detail to the statistical approach for these analyses and apologize for our previous lack of clarity. See the specific responses below for examples of each of these major changes, as well as added details tracked in the marked-up ms.*

Some more specific comments:

I agree with reviewer 1 that the title is a too vague. To me, the word "persists" implies that variation in resistance is maintained over some period of time whereas in fact there is resistance, genetic, and immunological variation that is correlated. It's also not immediately clear what "across scales" refers to.

- *The ms title has been changed to **'Risk alleles for tuberculosis infection associate with reduced immune reactivity in a wild mammalian host'***

1.203: I assume that q is the false discovery rate? If so, this should be explicitly stated, and some justification should be given for the two values that were chosen.

- *This has been clarified in lines 280-289. We also removed mention of the second significance value as it is not relevant to any of the figures or tables and is not referenced again. We now state a single significance threshold that aligns with the convention of 0.05.*

Figure 2: I might have missed something here but I was a bit surprised that the curves only start to drop off at around 4 years of age. Do younger buffalos not get infected at all?

- *We have clarified the initial age range for buffalo in the figure legend and added a sentence reminding the reader that animals positive at first capture were not included in the analysis, hence we would not be able to detect conversion in any animals younger than 2 and the first animal to convert to bTB positive out of this subset is about 3 years old.*

Referee: 1

Comments to the Author(s)

I would strongly suggest that the authors change the title to more clearly represent the findings of their study. The introduction lacks detail on the (very interesting) buffalo-bTB system, which is necessary to set the scene for the rest of the paper. For example, why is bTB important? The authors mention at the end of the MS, that it is “one of humankind’s most enduring scourges” but provide no evidence to back this statement up. The manuscript also lacks methodological detail in place (please see more detailed comments below). Finally, the discussion is very mechanistic in places, and (in its current form) more suited for a specialised audience.

- *Most of these comments are outlined below, but we have changed the title and added substantial detail to the introduction for bTB (lines 86-98). We have made an effort to more clearly explain the methods of the study and to cite out repetitive sections to other papers, as suggested below. We have also removed much of the technical language in the discussion, but not so much that it takes away from the tie between putative candidate genes and the cytokine mechanisms. We are happy to revisit any of these edits if the reviewers and AE should see fit. Please see specific examples below, as well as substantial changes to this section in the marked-up ms draft (e.g., lines 443-447, 461-468, and 470-501).*

I have included some more detailed comments below:

76 & 186: Please be clear about the findings of your previous study. I believe you didn’t actually detect heritable variation in this trait.

- *This finding has been clarified as marginal in the text (now lines 106-107 & 221-222).*

123: Please include the 95% confidence interval for this estimate of prevalence

- *CI has been added, now line 157.*

126: Here and throughout “INF” should be changed to “IFN”

- *Fixed in lines 160, 166, 335 and throughout.*

124 – 131: These references are all very general/not buffalo-specific. Are there any more specific studies that could be referenced here? If not, then it should be made clear that these statements are based on knowledge from other systems.

- *Now clarified as general patterns in cytokine function across mammals, now line 160.*

138: Please include more explanation re. stimulation with pokeweed i.e. testing potential of the immune system to respond to challenge.

- *Detail has been added to justify this proxy, now lines 173-177.*

140 – 145: These details should be included in a separate statistical section

- *Details were moved into the stats section for the cytokine models, now in lines 326-329.*

143: Why were some cytokine measures missing?

- *Missingness and additional detail about time series cytokine sampling is now explained in lines 185-192.*

173 – 175: This is unnecessary detail, I would suggest removing

- *Removed comparison of r^2 to D' as a metric of LD from line 214.*

159 – 62: This statement is too vague. Please include details.

- *The sentence referenced here was part of the section removed and cited out to Tavalire et al., 2018 so is no longer present.*

147 – 182: The majority of the genotyping methods are currently copied and pasted from the authors' previous paper (ref no. 30). I would suggest referring to this paper in places, to shorten this section/avoid repetition.

- *This section is now substantially shorter and cited out per the reviewers suggestion, retaining only the details needed for the PCA and GWAS analyses (now lines 193-211).*

261 – 263: These details are repeated (already mention on line 210), I would suggest removing one of these instances.

- *The details of out linkage block sliding window size are now cut form the methods but were retained in the results (now lines 380-382).*

197: Please include a sentence somewhere explicitly explaining why anti-helminthic treatment (though not the focus of this study) was accounted for in your models.

- *Now justified in lines 268-269.*

198: Were potential temporal effects accounted for in this model?

- *We have added additional detail to make it clear that only a single observation (age) for each animal was included in the Cox regression and that this was not a longitudinal mixed effects Cox regression (the use of 'age at onset' throughout and added detail in lines 266-274). We therefore don't believe that we need to account for temporal affects outside of a repeated-measures design, but are happy to discuss further if the reviewer wishes.*

198 & 225 – 227: Were fixed effects checked for collinearity? Please include details in MS.

- *We regret this oversight and this check is now documented in lines 278-281 for the Cox regression models and in lines 331-332 for the cytokine models.*

201: This is too vague. Please include more details e.g. what percentage of variation was explained by these 5 components?

- *Cumulative variation explained in now stated in lines 276-278.*

213: Please provide a brief description of how InterProScan works.

- *Added to lines 301-305.*

226: Please include a justification for the inclusion of these interaction terms

- *Justification is provided in lines 322-326.*

232 – 235: “Shapiro Wilk” not “Shapiro Wilks”. This statement is also unclear. Were models which violated the assumption of normality discarded? If so, how were models adjusted to ensure that residuals met this assumption? Were residuals also checked to ensure they were homoscedastic? Please include more detail here.

- *Now lines 338-349, language has been slightly altered to be more clear and additional detail for natural log transformations of cytokine data has been added.*

233: This statement re. model selection is also unclear. Elsewhere, you refer to “AIC-based selection”, please clarify.

- *This is now specific to the cytokine models (now line 347) and ‘AIC-based’ has been removed from line 399.*

284 – 287: You have not cited or discussed two relevant papers from a wild rodent system which show polymorphism at a specific gene associated with differences in the Th2 immune response and tolerance to macroparasites (Jackson et al. 2014 PLoS Biol 12; Wanelik et al. 2018 Mol Ecol 27).

- *Thank-you for pointing out these references. They are now discussed in lines 428-431. Additionally, we have clarified the language pertaining to novelty of our findings in the first sentences of the discussion (now lines 403-409).*

Referee: 2

Comments to the Author(s)

In this manuscript, the authors study risk of bovine tuberculosis (bTB) infection in populations of wild African buffalo, and identify two genetic loci in those populations that additively increase risk of bTB infection. The high risk allele at one of those loci correlates with reduced IL-12 production, suggesting lower macrophage activation. This fascinating study links genotypic variation in wild host populations to immune function (i.e., a putative mechanism) and risk of infection with a globally important disease. The manuscript is extremely well written, and I have only very minor comments.

Minor comments

L65: Typo: change “evidences” to “evidence”

- *Corrected, now line 68.*

L110-145: I found the methods presentation in the “bTB Testing and Cytokine Stimulations” section to be a little confusing. Specifically:

1. Were all of the blood samples (including the sample for bTB testing) from peripheral blood from the jugular vein? If not, from where on the buffalo did the blood for bTB testing come from?

- *This detail has been clarified in line 143.*

2. To help non-immunologists (like me...) understand the methods, I think it would be good if you used the word “cytokine” at least once in the paragraph in lines 124-131. That paragraph sets the stage for what variables you were interested in measuring, and why (which is great!), but

doesn't refer to any of the immune components as "cytokines". Then, the subsequent paragraph (lines 132-145) tells exactly how the interferon gamma, IL-12, and IL-4 were measured, but only refers to those components as "cytokines." I am embarrassed to say that I had to Google the word "cytokine" just to confirm that the interferon gamma, interleukin 12, and interleukin 4 were indeed what lines 132-145 were talking about.

- *We apologize for the confusion and inconsistency in language- we have now clarified that we are talking about measuring cytokines in this paragraph in lines 158-159 and 167. We also added detail to guide the non-immunologist through this paragraph in lines 161-167.*

L117-122: Just to be clear, you obtained time series of 2-9 bTB tests for each animal, but only animals with time series of 4-9 bTB tests were included in the analysis (i.e., animals with fewer than 4 test results were excluded)? If so, I feel this could be stated a little more clearly.

- *In response to this comment, inclusion and exclusion criteria have been more clearly outlined now in lines 150-156.*

L159: Typo: Alf1 is written AlfI here.

- *This detail is now part of the methods cut and cited out to Tavalire et al., 2018.*

L252: I think this sentence would be improved by changing "though" to "and" ("These two loci had an additive effect on bTB conversion risk, *and* no interactive effects were detected.")

- *This change has been made- now lines 366-367.*

Appendix B

We thank the Associate Editor Board Member, one original reviewer, and one additional reviewer for their thoughtful feedback on this second draft of the manuscript entitled “Risk alleles for tuberculosis infection associate with reduced immune reactivity in a wild mammalian host.” We believe these comments have strengthened our submission and are happy to discuss any issues the AE or reviewers may have. Please find our responses below indented and in italics. Line numbers correspond to the clean version of the revision.

Associate Editor Board Member

Comments to Author:

The authors have done an excellent job at addressing the reviewers' comments on their previous submission. The paper has now also been reviewed by a 3rd reviewer who is also very positive. There are some outstanding comments from both reviewers 1 and 3 that would need to be addressed but most of these comments relate to improving the clarity of presentation and should therefore be relatively straightforward to address.

I agree with reviewer 3's suggestion of further improving the introduction. As it stands, statements on general immunological principles, immunology of Mycobacterium and epidemiological statements on Mycobacterium are all intermingled within paragraphs. It would be good to better separate those statements and the structure suggested by Reviewer 3 seems suitable to achieve this.

- You will find the introduction in the current draft largely restructured per reviewer 3's helpful insight. Please see detail below in our response to that comment.

Another, very minor comment about the Introduction: I found the statement in lines 78-79 that M. bovis makes up a larger fraction of human TB than previously estimated very interesting but felt it was rather vague. It would be good if the authors could provide some actual numbers here. (It makes a big difference whether previous estimates were 0% and now they are 5% or whether they were previously 0.001% and now they are 0.0015%!)

- Now lines 53-56: We appreciate this request and have added numeric context to this sentence.*

Reviewer(s)' Comments to Author:

Referee: 1

Comments to the Author(s).

I am very happy with the authors' thorough responses to my concerns, and believe the paper is much improved. I have two outstanding comments:

(1) My original comment (line 198) referred to the possibility of including a year effect in the Cox regression model i.e. which year an individual was sampled in. Could this affect conversion risk?

- We apologize for our initial misinterpretation of this comment and have taken this into full consideration. We resolved this issue by including initial year captured in the full Cox PH regression model for the two loci of interest, however this term drops out during AIC-based model selection and is not included in the final model. The initial inclusion of this term as a covariate is now detailed in lines 244-247.

(2) I am still slightly concerned about the authors' representation of the novelty of this study. I would suggest changing lines 330-331 to read: "This is one of very few studies taking a multi-level approach, connecting disease phenotypes across scales, from an individual-level disease trait to patterns in immune reactivity in the wild."

- Now lines 347-349: This sentence has been reworded as suggested.

Referee: 3

Comments to the Author(s).

The authors present the results of an innovative study, in which they link findings from the genome- to the population-level to identify genes that encode for elevated risk of positive infection with Mycobacterium tuberculosis in a wild African buffalo study system. The authors convincingly demonstrate that buffalo possessing one of two single nucleotide polymorphisms (SNPs) identified in their analysis are statistically more likely to contract Tb at an earlier age than those lacking the polymorphism. These genes appear to be additive, meaning that those individuals positive for both SNPs are at the greatest infection hazard of all. The authors link these genotypes to immune phenotype by demonstrating that individuals positive for one of their self-identified "risk alleles" demonstrate reduced production of IL-12, an important cytokine in phagocyte activation and anti-Tb defenses, across their lifetimes, suggesting a mechanistic basis for reduced resistance to infection. The authors further show that these genome

variations are not associated with any variation in lifetime production of IFN-gamma or IL-4, indicating no SNP-associated variation in Th1 or Th2-primed immunity. Rather, these genetic anomalies appear to be linked to IL-12 and anti-Tb defenses specifically.

In general, I find this to be a compelling and fascinating study that is noble in its efforts to bridge between within-host immunological mechanisms and population-level ecological patterns. I have a few concerns with a lack of precision in some of the language, especially in the introduction, concerning definitions of the terms “resistance” and “tolerance” as well as “disease.” In particular, the authors highlight high population-level maintenance of Tb in the African buffalo system (lines 88-97) but then emphasize resistance alleles that would limit infection rates for these host. Please resolve these inconsistencies or discuss how variation in individual resistance might contribute to high population-level prevalence (perhaps by restricting infection to later age individuals and reducing lifelong mortality effects). Re: use of the term “disease,” I would swap this for “infection” when possible to avoid needing to demonstrate some sort of pathological manifestation in the host. The authors present no information on the morbidity or mortality effects of Tb infection for buffalo (though see comments on Discussion for how some of this might be helpful to understand evolutionary pressures in this section).

- We thank this third reviewer for their thoughtful insight. The concerns listed above are further detailed below, but briefly:

- Re: definitions: Tolerance and resistance are now explicitly defined in the intro on lines 23 and 27 respectively. The word ‘disease’ has been changed to ‘infection’ throughout, wherever possible.

- Re: population versus individual-level dynamics: We have tried to clarify that resistance operates on a continuum in this system and have added discussion of how age-dependent infection patterns and fitness effects might drive high prevalence at the population level in lines 84-99.

- Re: mortality: information has now been added to the intro and discussion on mortality once infected (lines 93-97, see below for details on discussion).

Additionally, I think the intro could be better organized. As outlined below, I suggest restructuring the paragraphs into the following order: (a) intro to the rarity of cross-scale studies, (b) intro the global burden of Mycobacterium across species, including humans and cattle, (c) intro to Tb’s methods of immune avoidance, (d) highlight how buffalo are effective at resisting infection nonetheless, suggesting some heritable resistance traits (but be careful to follow precise definitions of resistance and tolerance), and (e) use this to motivate the imperative in understanding resistance mechanisms against this pathogen so as to understand infection risks in other hosts.

- *We thank you for this helpful restructuring feedback and have restructured the sections of the introduction per your suggestion and have added some text as well.*

Specific comments:

Line numbers refer to the clean, corrected draft in the absence of track changes.

Abstract:

Line 4: Suggest adding commas around “along with environmental factors” to eliminate confusion from sentence structure

- *Now line 4: commas have been added.*

Line 7: Suggestion to drop the term “transmission” as it is not often observable in the wild but usually inferred via model fitting—and it is not actually observed here. In this study, we witness evidence of infection conversion but the actual transmission event is not demonstrated (and is, to my knowledge, quite poorly understood). Given the authors’ desire to highlight something that is “uniquely observable in natural populations”, I suggest the following rewording:

“However, genetics and immunology are typically studied in model systems, whereas population-level patterns of infection status and susceptibility are uniquely observable in nature.”

- *Now lines 5-7: This sentence has been reworded as suggested.*

Also, I’m not a huge fan of the term “disease” as it carries so many implications about effects on host health.

- *‘Disease’ has been changed to ‘infection’ wherever possible, and here in line 15 (see mark-up, below).*

Line 11: Keep to present tense to be consistent with the rest of the abstract

- *Now line 12: Fixed.*

Intro:

Again, reiterating my objection to the term “disease” or at least a request that it be crisply defined...

- *Fixed throughout.*

Line 23: you need to define what you mean by ‘tolerant’ since it is a loaded term. I’m assuming this is the ecological definition, meaning that hosts carry high pathogen burdens but that their health is not impacted?

- *Now defined in line 23.*

Line 48: Would be good to define “effector cell” here, since the paper claims to be targeting a wide range of sub-disciplines for readership

- *With the restructuring, this section has been cut from the MS; these cells are not referred to as immune cells or innate immune cells.*

Line 52: How are pathogen-recognition mechanisms integral to pathogen survival? Please clarify what you mean here.

- *Now line 66: We apologize for the sentence structure creating confusion, we have changed this sentence to be more clear that the pathogen proteins are integral to pathogen survival.*

Line 68: Should be two sentences here. Also, the motivation for the study is a bit weak—why do you look for resistance mechanisms if limited evidence of this exists?

- *Now line 78: Broken into two sentences now. Also lines 78-83 have been reworded to clarify strength of the motivation.*

Suggestion to restructure the whole intro section as follows: (a) intro cross-scale studies as done here, (b) intro the burden of *Mycobacterium* across species and global human population, (c) highlight its crafty methods of immune avoidance, (d) but highlight how buffalo are effective at resisting it nonetheless, and (e) use this to motivate the imperative in understanding resistance mechanisms against this pathogen so as to alleviate disease burden in other hosts.

- *This comment has immensely improved the intro section. Please see tracked changes below for all changes. Each section outlined above starts on lines: (a) 19, (b) 48, (c) 62 (with reduced general immunology text in this section), (d) 84, and (e) 101.*

Also, be careful with terms – you talk a lot about “resistance loci” but then say that buffalo sustain high levels of infection... It is certainly possible for both resistance and tolerance to be at play, but you need to make sure you use your terms correctly and carefully.

- *Heterogeneity in resistance has been clarified in the last paragraph (lines 90-97).*

Methods

In general, the methods are described thoroughly, succinctly, and clearly. As I am not a population geneticist myself, I cannot assess SNP Genotyping and Filtering methods described in this paper and suggest to the editor that at least one reviewer in this field should comment on the manuscript.

Line 250: spp. should not be italicized here.

- *Now line 265: Corrected.*

Results

Line 313 and Figure 3: Figure 1 suggests that animals possessing both “G_” and “T_” risk alleles are at a compounded risk of obtaining positive Tb infection, over and above those heterozygous for each. Can you include separate curves for each of these genotypes in Fig 3 as well, so that we can visualize allele-specific differences in effects on immune function?

- *We appreciate this comment, however since alleles at SNP3195 were not selected in the final, best fit models for cytokine production, we hesitate to add this to our groupings in a main figure, since this would obscure our results and potentially confuse the reader relative to table 3. However, we have created a figure with all genotypes, as suggested here, and added it to the supplement as figure S2. We have also altered the colors in figure 2 to match the supplementary figure as well to improve clarity.*

From your mixed effects model (Table 3), it appears that only the “G_” allele is significantly associated with reduced IL-12 production. In discussion of these results in the text, you do not address the absence of relationship for the “T_” allele with cytokine production patterns. Please report these results and include a line or two in the Discussion speculating on the mechanism of this gene’s effects on Tb susceptibility.

- *Though we did report these results in the original draft, and here, we have expanded upon this description, as suggested, in lines 335-341. Additionally, we have added some discussion of SNP3195 to the discussion in lines 358-362, however we hesitate to name putative mechanisms, as we could not annotate this region of the genome (this is now more clearly stated).*

Line 316: I’m not sure how Fig 3 shows that IL-12 is not affected by Tb status. Suggestion to remove this reference and include only the ref to Table 3 (which demonstrates this lack of association clearly).

- *Now line 331: We apologize for this oversight and have removed the reference to fig 3.*

Discussion

In general, discussion is clear and concise. As mentioned above, I'd like to see some discussion of the interactive effects of the two SNPs identified on infection risk and the absence of effect for the second allele on cytokine production. Are there other immune factors that the authors did not assay that could underly the mechanism of this genotype's effects on infection risk?

- *Clarified in lines 358-362, however we hesitate to name putative mechanisms, as we could not annotate this region of the genome (this is now more clearly stated).*

Additionally, some discussion of the evolutionary underpinnings and maintenance of the identified SNPs would be helpful. How are these SNPs maintained? What is the mortality hazard imposed by Tb infection on African buffalo? The authors mention in the introduction that high infection rates are common throughout wild populations but do not indicate whether those positive for infection are also more likely to die early. It would be good to have at least some sense of the evolutionary selective environment in which these SNPs have arisen and been maintained.

- *We have now added a discussion of the selective environment in lines 381-390.*